# Vertical Stress Induced Anomalous Spectral Shift of 13.17° Moiré Superlattice in Twist Bilayer Graphene

**DOI:** 10.3390/molecules28073015

**Published:** 2023-03-28

**Authors:** Wenjing Miao, Hao Sheng, Jingang Wang

**Affiliations:** Liaoning Provincial Key Laboratory of Novel Micro-Nano Functional Materials, College of Science, Liaoning Petrochemical University, Fushun 113001, China; miaowenjing@stu.lnpu.edu.cn (W.M.); jingang_wang@lnpu.edu.cn (J.W.)

**Keywords:** twist bilayer graphene, vertical stress, Bloch wave function, charge density difference, absorption spectrum

## Abstract

The electronic states of the twist bilayer graphene (TBG) moiré superlattice are usually regulated by the rotation angle, applied electric field, applied magnetic field, carrier concentration and applied stress, and thus exhibit novel physical properties. Squeezing, that is, applying vertical compressive stress to the graphene layers, has profound significance in regulating the photoelectric properties of the moiré superlattice and constructing optical nanodevices. This paper presents the photoelectric properties of a TBG moiré superlattice with a twist angle of 13.17° and tunability under vertical stress. Interlayer distance decreases nonlinearly with compressive stress from 0 to 10 GPa, giving rise to weakened interlayer coupling compared to a Bernal-stacked graphene bilayer and an enhanced repulsive effect between the layers. The calculated Bloch wave functions show a strong dependence on stress. With the increase in stress, the band gaps of the system present a nonlinear increase, which induces and enhances the interlayer charge transfer and leads to the redshift of the absorption spectrum of the moiré superlattice system. By analyzing the differences in the Bloch wave function and charge density differences, we explain the nature of the physical mechanism of photoelectric property change in a stress-regulated twist superlattice system. This study provides a theoretical basis for the identification of piezoelectric properties and the stress regulation of photoelectric devices based on TBG, and also provides a feasible method for regulating the performance of TBG.

## 1. Introduction

Twist bilayer graphene (TBG), which is double-layer graphene with a twist angle, forms a periodic moiré superlattice (MS) due to structural changes between the layers [1,2,3]. TBG, due to the reconstruction of the C atom, has novel physical properties such as the strong correlation effect of interlayer electrons [4,5], a topological protection state [6,7,8], spontaneous ferromagnetism [9,10,11], a Mott insulator [12,13,14], etc., which have become the focus of nanomaterials [15,16,17].

The transition from an insulator to a superconductor can be achieved by tuning the carrier concentration in TBG [18,19]. In addition, by changing the twist angle, the Fermi velocity in the MS can be changed to effectively tune the van Hove singularity (VHS) [20,21,22,23], thereby tuning the optical and electrical properties of the TBG [23,24,25,26,27,28]. A large number of studies has shown that it is effective to adjust the optoelectronic properties of TBGs by means of changing the twist angle [29], external electric field [30], magnetic field, carrier concentration and so on [31,32,33,34,35,36]. However, the twist angle in experiments between a Mott insulator and superconductor has to be distorted to about 1.1°, which is extremely challenging, Additionally, very small permutation errors can make all the difference. The results show that the electrical properties and phase of TBG can be regulated by applying axial pressure to TBG at any twist angle [37]. The pressure makes it easier to tune the band structure of the TBG superlattice [37]. Experimental studies have shown that under the action of axial strain, the energy band structure of nanomaterials will change, resulting in novel physical properties such as solitons and photonic crystals, and in research on piezoelectric materials and devices.

In this work, we focus on the regulation of TBG spectral properties by adjusting pressure. Firstly, the energy band structure and the Bloch wave function of periodic MS were studied by theoretical calculation. Further, the distribution of electrons/holes in MS was analyzed. Corresponding to the vertical stress, the absorption spectrum of MS exhibits regular changes. All these provide the theoretical basis for the application of TBG in the optoelectronic and piezoelectric fields. The theoretical results can guide the precise fabrication of optical devices with special properties, thereby promoting the application of MS in optical spectroscopy devices.

## 2. Results and Discussion

A schematic diagram of the TBG moiré superlattice with a twist angle of 13.17° is given in Figure 1a. Figure 1b shows that the interlayer distance decreases from 3.38 Å to 2.81 Å when the vertical stress ranges from 0 GPa to 10 GPa. When stress applied increases from 1 GPa, the distance between layers decreases sharply. Additionally, then, the decreasing rate gradually gets smaller with increasing vertical stress, which is mainly due to the increasing repulsive forces between the C atoms in the layers. In order to monitor the change in interlayer distance with the stress, we took the averaged ordinate of the underlying graphene layer as the reference and plotted the spatially distributed Z coordinates of the upper graphene layer under the stress of 0 GPa, 4 GPa, 7 GPa, and 10 GPa, as shown in Figure 1c–f. Obviously, the spacing value between two layers, as given in the color bar, decreases with stress. The AA stacking area of TBG of a bright red or yellow color has larger spacing than the AB and BA area, which is also observed in other 2D materials [38]. However, the spacing difference between the AA and AB (or BA) areas is strongly regulated by stress, and becomes negligible as stress goes beyond 4 GPa, as seen in Figure 1c–f. By fitting the stress and strain curves, as seen in Figure 1b, we deduced that the interlayer coupling K coefficient of TBG in this paper is 48 × 10^18^ N/m^3^, about half of the value of K = 106.5 × 10^18^ N/m^3^ [39] of the AB (Bernal)-stacked graphene bilayers, which indicates that twist can effectively tune the interlayer coupling strength. The calculated Poisson’s ratio is 1.3, which is unusually large but has a larger fitting error.

The direct consequence of the change in interlayer distance is the change in the band gaps of TBG at the K and M points (Figure 2). The zoomed-in plot of the K-point band gaps, as given in Figure 2b, shows that as the vertical pressure changes from 0 GPa to 1 GPa, the K-point band gaps decreases slightly, and then increases gradually with the increase in vertical stress. It indicates that the metallic properties of the TBG superlattice system become weaker and the insulation is strengthened under vertical stress. Figure 1c,d shows more clearly the nonlinear dependence of the band gaps on vertical stress for the K and M points, respectively. Different from the K-point, the band gaps at the M point decrease with increasing stress. The opposite behavior between the M-point and K-point gaps can actually be comprehended from the stress-induced bandwidth decrease.

In order to investigate the reason why the band gaps are regulated by vertical stress, the Bloch wave function was analyzed. (Figure 3) The study showed that the Bloch wave function of TBG is affected. The wave function for both interband and intraband transitions shows that the amplitude of the wave function increases immediately when the vertical stress increases from 0 GPa to 1 GPa, and then gradually increases with the increase in the vertical pressure from 1 GPa to 10 GPa. However, when the vertical stress increases from 0 GPa to 1 GPa, the amplitude of the wave function of the interband transition increases more obviously, especially in the region without MS, which is consistent with the results of previous studies.

The change in the Bloch wave function then causes the separation behavior of electron and hole in the transfer process to be significantly different, as shown in Figure 4. Figure 4a,c shows charge density difference (CDD) for interband transitions and intra-band transitions at vertical stresses of 0 GPa, 1 GPa, 5 GPa and 10 GPa, respectively. With the increase in stress, the degree of charge transfer gradually increases. It is obvious that in the interband transfer process, due to the low energy, the change in CDD is very obvious when the vertical stress increases from 0 GPa to 1 GPa, and then increases gradually with the increase in vertical stress above 1 GPa. When the stress is greater than 8 GPa, the CDD of TBG gradually decreases. On the other hand, the energy in the intraband transfer process is high, so the CDD does not change much from 0 GPa to 1 GPa. After the charge transfer induced by stress increases, the interlayer charge transfer is enhanced. This indicates that the system has an inflection point when the vertical stress increases to 8 GPa, which has practical significance for the application of TBG in the piezoelectric field.

Stress regulates optical absorption properties by inducing charge transfer (Figure 5). It is not difficult to see that with the increase in vertical stress, the optical absorption of TBG appears as the phenomenon of red shift, which is due to the enhanced charge transfer between layers and also the decreased bandwidth (as indicated in Figure 2d) induced by stress. The absorption intensity changes abruptly as the stress changes from 0 GPa to 1 GPa. However, in the NIR, the absorption intensity decreases to that of no stress at 8 GPa. This phenomenon can be traced back to the previous result that charge transfer is regulated by pressure (Figure 4d). On the other hand, the theoretical study of the dielectric function shows that the TBG superlattice has an obvious plasmon effect in the light wave region of 1100 nm~1900 nm (blue region in Figure 5c; the real part is negative and the imaginary part is positive), and the plasmon effect gradually decreases with the increase in stress. In addition, the larger the value of vertical stress and the longer the wavelength, the more obvious the red-shift phenomenon of the absorption spectrum, which is because the Bloch wave function of TBG is induced by stress. The closer it is to the shallow Fermi level, the more susceptible it is to pressure. Because the lattice is less susceptible to deformation due to stress, only shallow energy levels such as HOMO and LUMO are improved.

## 3. Materials and Methods

The QuantumATK−2019 program package (fermitech, Beijing, China) was used to optimize the structure of TBG and calculate the band structure [40], the density of states and the optical properties. Among them, the basis set was LCAO (linear combination of atomic orbitals), pseudo potentials were determined by Pseudo Dojo [41] and electronic exchange-correlation potential was treated by the generalized gradient approximation functional of Perdew Burke Ernzerh (GGA-PBE), combined with the density functional theory with Grimme’s D3 correction and without damp (DFT-D3) dispersion correction [42]. For calculation accuracy, the energy cutoff of LCAO was 800 eV, and the K-mesh was 2 × 2 × 1.

## 4. Conclusions

In this work, we used the first principles method to theoretically calculate the electronic structure and optical properties of twist bilayer graphene (TBG) under the control of vertical stress. It was shown that the distance between TBG superlattices decreases nonlinearly with increasing vertical stress from 0 GPa to 10 GPa at the twist angle of 13.17°, and that the decreasing rate gets smaller with increasing stress. We deduced the interlayer coupling K coefficient of TBG in this paper, which is approximately 48 × 10^18^ N/m^3^, which is almost half as much as that in AB-stacked bilayer graphene rendering twist a means to effectively tune interlayer interaction. With the increase in stress, the band gaps in the band structure of a TBG superlattice system increases, the metallic properties of TBG superlattice system are weakened and the insulation is enhanced. The amplitude of the wave function increases faster and then slower for interband transition and intraband transitions, and the turning point is at 1 GPa. The interlayer charge transfer is induced by the change in band gaps, and the CDD of TBG increases gradually when the stress ranges from 0 GPa to 8 GPa. When the stress is greater than 8 GPa, the CDD of TBG gradually decreases. The interlayer charge transfer of the system results in the red shift of the absorption spectrum of the moiré superlattice system. When the applied stress is 1 GPa, the absorption peak increases significantly, and then decreases gradually with the increase in stress. When the wavelength range is 1300–2100 nm and the vertical stress is 8 GPa, the absorption strength is the same as when there is no stress. In the range of visible light, the optical absorption coefficient of 1–10 GPa under external stress is always greater than that without stress. In the visible range and near-infrared range, the optical absorption of TBG appears as a redshift with the increase in vertical stress. The theoretical study of dielectric function showed that the plasma effect of the TBG superlattice at 1100–1900 nm gradually decreases with the increase in stress. Our study explains the physical mechanism, and photoelectric and piezoelectric properties of the TBG superlattice under vertical pressure. At the same time, it also provides a research basis for the theoretical research and application of TBG superlattices in photoelectric and piezoelectric sensors.

## Figures and Tables

**Figure 1 molecules-28-03015-f001:**
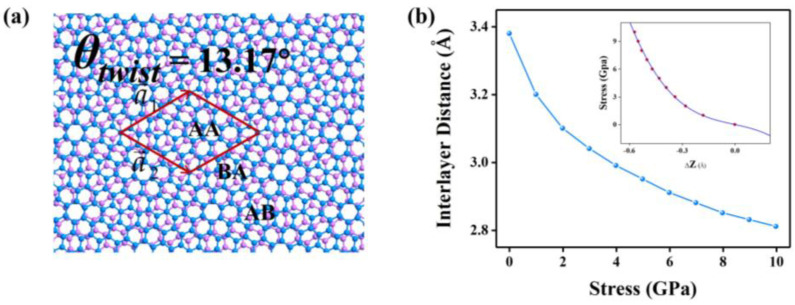
(**a**) TBG moiré superlattice with a twist angle of 13.17°. (**b**) Relationship between vertical stress and interlayer distance. Inset shows the fitting curve between stress and strain. (**c**–**f**) The spatial distribution of interlayer distance as a function of compressive stress values of 0 GPa (**c**), 4 GPa (**d**), 7 GPa (**e**), and 10 GPa (**f**). The interlayer distance is defined as the Z coordinates of the upper graphene layer with respect to the averaged Z coordinate of the lower layer. The stacking modes of AA, AB and BA in (**c**–**d**) correspond to those in (**a**).

**Figure 2 molecules-28-03015-f002:**
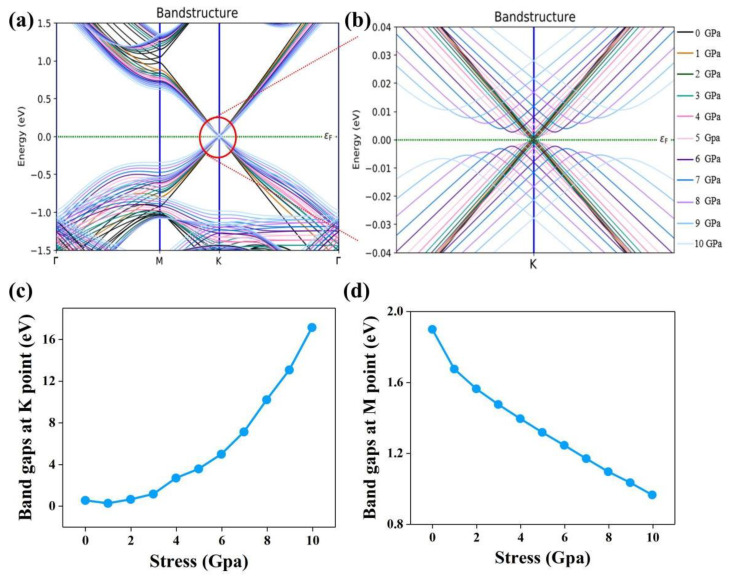
(**a**) The band structure of TBG as a function of vertical stress. (**b**) The zoomed-in band structure from the area circled in red in (**a**). Stress-dependent band gaps at point K (**c**) and at point M (**d**).

**Figure 3 molecules-28-03015-f003:**
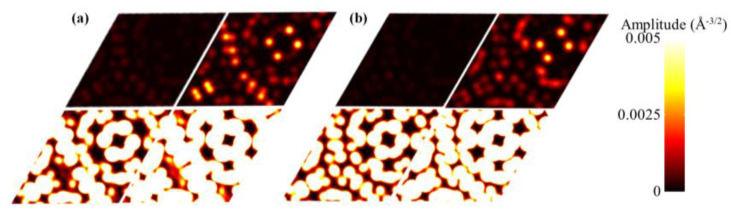
Colorization plots of wave functions for unit cell’s interband transitions (**a**) and intraband transitions (**b**) of superlattices with vertical stresses of 0 GPa, 1 GPa, 5 GPa and 10 GPa, respectively.

**Figure 4 molecules-28-03015-f004:**
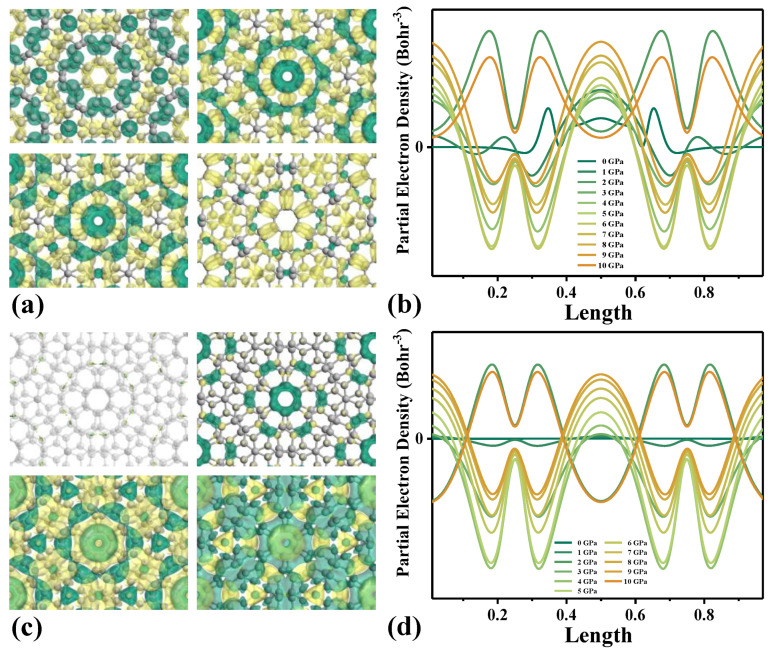
CDD of interband transition (**a**) and intraband transition (**c**). Partial electron density of interband transition (**b**) and intraband transition (**d**).

**Figure 5 molecules-28-03015-f005:**
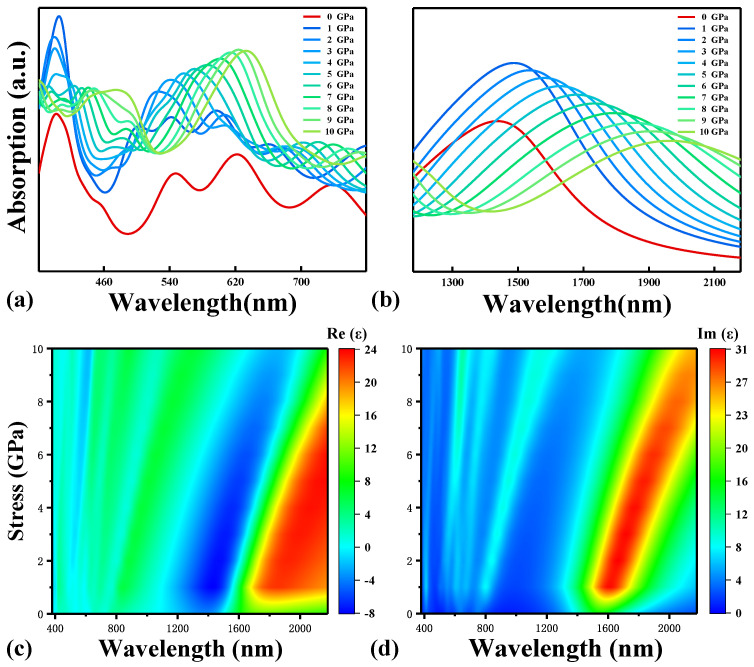
Absorption spectrum of the visible light wave range (**a**) and near infrared light wave range (**b**). The real (**c**) and imaginary (**d**) parts of the dielectric function.

## Data Availability

Data is unavailable due to privacy or ethical restrictions.

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
