# Peer review of "Vertical Stress Induced Anomalous Spectral Shift of 13.17° Moiré Superlattice in Twist Bilayer Graphene"

_molecules, 2023, doi:10.3390/molecules28073015_

Round 1

Reviewer 1 Report

TBG is one of the booming fields in recent years and its potential applications have been intensively explored. Vertical stress effect on the electronic, optical properties of TBG is important and studied by using density functional calculations for a possible opto-electronic devices in this paper. Usually effective continuum model was routinely used to calculate electronic band structure due to a huge number of atoms in unit cell in small twist angle system. The authors carefully chose a specific twist to make a DFT calculation possible, which is good and can help test model Hamiltonian calculation. The paper can be considered for publication in the journal, but the following questions have to be clarified,

(1) The vertical compressive stress is applied, and an important issue in the paper. I wonder if the Poisson ratio is considered, since the in-plane lattice size will be changed (increase) accordingly. I made a deduction (see attached pdf) of vertical stress as a function of strain, and found that there should be a nonlinear relation between stress and strain, as also indicated in Figure 1a in the paper. Please add some more details in the method part on the structural optimization with the Poisson ratio considered, if such an optimization was performed. If not, please consider such an effect in the paper. 

(2) From Figure 1d, along with the increasing compressive stress, band gap increases. Is it a direct gap or indirect one?

(3) From Figure 4, the optical absorption peaks shift to larger wavelength (or smaller band gap) with the increasing stress, it seems not consistent with the result in Figure 1d? We understand that the authors are more interested in the visible spectrum, but in this high-frequency range the absorption data is not directly related to the band gap data in Figure 1. The band gap size in Figure 1d is around 10-20 meV (figure 1d), or 62-124 micrometers in wavelength, but this wavelength range is not shown in Figure 4.

Shall we understand that in the long-wavelength range (62-124 micrometers in the near and far infrared spectrum), the optical absorption peaks shift in an opposite trend to that in visible range, namely, peaks shifting to larger energy (shorter wavelength) when stress increases? Please explain this a little bit.    

(4) Could the authors give some more structural information if not too difficult, for example, the z-component distribution of all the atomic positions and its dependence on the compressive strain in the optimized structures? We raise up this issue, because in the model Hamiltonian calculation, such interlayer parameter is usually unknown but important, and should be taken into account for a more precise simulation. And this information, if supplied in DFT calculations, is nice for the community of twistronics to study the small-twist systems.   

Reviewer 2 Report

see attached file

Round 2

Reviewer 1 Report

Obviously the authors have made some good efforts to improve the quality of the presentation and obtained some encouraging results (such as interlayer coupling strength and Poissons ratio).

When reading the reply letter, I just realized that I have an answer to one of my questions in the previous reviewing report (the inconsistency between band gap and optical absorption with stress). It is actually already shown in Figure 1b. From Figure 1b, the band gap in the K point increases with compressive stress, but meanwhile, the band gaps in the area of the Gamma and M points, which lies in the range of visible light, decreases. This can explain well the stress-induced redshift in the absorption data in Figure 4.

The decreasing band gap in G and M and increasing band gap in K is also easy to understand, most likely due to the shrinked band width due to the in-plane elongation induced by out-of-plane compressive stress (Poisson's ratio).

I carefully read the new version of the manuscript and made some suggested modifcation in the attached PDF file.

Technical issues:

New Figure 2. The values in the color bar are the same, please choose a proper range of data for (b-d).

With all these incorporated, I would like to recommend it for publication in Molecules journal.
